

# Penetrance of symbiont-mediated parthenogenesis is driven by reproductive rate in a parasitoid wasp

Amelia R.I. Lindsey and  Richard Stouthamer

Department of Entomology, University of California, Riverside, Riverside, CA, United States of America

## ABSTRACT

*Trichogramma* wasps are tiny parasitoids of lepidopteran eggs, used extensively for biological control. They are often infected with the bacterial symbiont *Wolbachia,* which converts *Trichogramma* to an asexual mode of reproduction, whereby females develop from unfertilized eggs. However, this *Wolbachia*-induced parthenogenesis is not always complete, and previous studies have noted that infected females will produce occasional males in the lab. The conditions that reduce penetrance of the parthenogenesis phenotype are not well understood. We hypothesized that more ecologically relevant conditions of limited host access will sustain female-biased sex ratios. After restricting access to host eggs, we found a strong relationship between reproductive rate and sex ratio. By limiting reproduction to one hour a day, wasps could sustain up to 100% effective parthenogenesis for one week, with no significant impact on total fecundity. Reproductive output in the first 24-hours appears to be critical to the total sex ratio of the entire brood. Limiting oviposition in that period resulted in more effective parthenogenesis after one week, again without any significant impact on total fecundity. Our data suggest that this phenomenon may be due to the depletion of *Wolbachia* when oviposition occurs continuously, whereas *Wolbachia* titers may recover when offspring production is limited. In addition to the potential to improve mass rearing of *Trichogramma* for biological control, findings from this study help elucidate the context-dependent nature of a pervasive symbiotic relationship.

# INTRODUCTION

*Wolbachia* is a maternally transmitted, symbiotic bacterium that inhabits numerous arthropods and nematodes. Its ubiquity can be attributed to both fitness advantages for the host, and reproductive modifications of the host. Known reproductive modifications include cytoplasmic incompatibility, male-killing, feminization, and parthenogenesis-induction (*Werren, Baldo & Clark, 2008*), all of which increase the relative fitness of infected females, thus allowing *Wolbachia* to spread through a population (*Hoffmann et al., 2011*; *Turelli & Hoffmann, 1991*). Parthenogenesis-inducing *Wolbachia* infect haplodiploid species and result in the production of females without the need for a mate. This is accomplished through converting unfertilized eggs (that would otherwise develop as males) to diploid eggs, which then develop as females (*Gottlieb et al., 2002*; *Pannebakker et al., 2004*; *Stouthamer & Kazmer, 1994*).

Corresponding author
Richard Stouthamer,
richard.stouthamer@ucr.edu

There is a large body of research indicating that the phenotypes *Wolbachia* induces are context dependent, with a range of genetic and environmental factors influencing the penetrance of the manipulation (proportion of individuals displaying the phenotype) (*Mouton et al., 2007*; *Pascal et al., 2004*; *Serbus et al., 2015*; *Wiwatanaratanabutr & Grandjean, 2016*). These are important considerations for several reasons. Firstly, with symbionts under exploration for the control of target pest species (*Bourtzis et al., 2014*; *Hoffmann et al., 2011*; *Hoffmann, Ross & Rasic, 2015*; *Walker et al., 2011*), it is critical that we understand the dynamics that result in the desired host-symbiont extended phenotype, and the persistence of the infection in the target population. Secondly, the persistence of a symbiont in a host population, and expression of resulting phenotypes will affect the potential for host-symbiont co-evolution. The levels of maternal transmission, penetrance of the reproductive modification or manipulation, relative fitness costs or benefits for the host, and the proportion of infected individuals in the population all play into the ability of *Wolbachia* to spread and maintain itself in a population (*Hoffmann et al., 2011*; *Hoffmann, Turelli & Harshman, 1990*; *Turelli & Hoffmann, 1995*).

Changes in host genotype or the introduction to a novel host can result in altered *Wolbachia* titers (*Mouton et al., 2007*; *Watanabe, Kageyama & Miura, 2013*), failure to induce the anticipated phenotype (*Bordenstein, Uy & Werren, 2003*; *Grenier et al., 1998*; *Huigens et al., 2004*; *McGraw et al., 2001*; *Reynolds, Thomson & Hoffmann, 2003*), reduced maternal transmission, and the eventual loss of the symbiont from a population (*Huigens et al., 2004*; *Newton, Savytskyy & Sheehan, 2015*). Additionally, there are well-characterized relationships between several environmental factors and the penetrance of *Wolbachia*-mediated phenotypes. High temperatures will reduce *Wolbachia* titers and result in poor host manipulation (*Bordenstein & Bordenstein, 2011*; *Hurst et al., 2000*; *Pascal et al., 2004*). The same result has been found for antibiotic treatments: the higher the antibiotic dose, the lower the symbiont titer, and the lower the penetrance of the reproductive manipulation (*Zchori-Fein, Gottlieb & Coll, 2000*). In the case of cytoplasmic incompatibility-inducing *Wolbachia*, this means heat-treated male offspring of are incapable of inducing cytoplasmic incompatibility, or only do so weakly (*Clancy & Hoffmann, 1998*). In the case of parthenogenesis-inducing *Wolbachia*, antibiotic treated mothers produce increasingly more sons at lower *Wolbachia* titers (*Stouthamer & Mak, 2002*; *Zchori-Fein, Gottlieb & Coll, 2000*). Many of these studies point to a "threshold" level of infection that is critical for host-manipulation (*Bordenstein & Bordenstein, 2011*; *Hurst et al., 2000*; *Ma et al., 2015*), and a positive correlation between *Wolbachia* titers and expression of the manipulation (*Bourtzis et al., 1996*; *Breeuwer & Werren, 1993*; *Ikeda, Ishikawa & Sasaki, 2003*; *Pascal et al., 2004*; *Zchori-Fein, Gottlieb & Coll, 2000*).

The effect of removing or reducing *Wolbachia* titers is well documented for wasps in the genus *Trichogramma* (*Pintureau, Chapelle & Delobel, 1999*; *Stouthamer, Luck & Hamilton, 1990*; *Tulgetske & Stouthamer, 2012*). *Trichogramma* are minute parasitoid wasps in the superfamily Chalcidoidea, frequently infected with parthenogenesis-inducing *Wolbachia* (*Stouthamer et al., 1993*; *Stouthamer, Luck & Hamilton, 1990*; *Stouthamer et al., 1990*). Like other hymenopterans, *Trichogramma* are haplodiploid: unfertilized eggs typically develop into males, and fertilized eggs into females (*Stouthamer, Luck & Hamilton, 1990*).

In *Trichogramma*, parthenogenesis-inducing *Wolbachia* restores diploidy of unfertilized eggs through via a failed anaphase in which chromosomes do not separate during the egg's first mitotic division (*Stouthamer & Kazmer, 1994*). For *Trichogramma*, increased doses of heat reduce bacterial titers and lead to the production of increasingly more males and sexually aberrant individuals (*Pascal et al., 2004*; *Stouthamer, 1997*; *Tulgetske & Stouthamer, 2012*). It is not clear however, why males are occasionally produced in the absence of antibiotics or increased temperature regimes (*Hohmann, Luck & Stouthamer, 2001*; *Stouthamer & Luck, 1993*).

The production of males could be useful in determining what factors control the expression of the symbiont phenotype. Preliminary studies that suggest that limited access to host eggs results in more female-biased sex ratios (*Hohmann, Luck & Stouthamer, 2001*; *Legner, 1985*; *Stouthamer & Luck, 1993*). However, the relationship between access to host eggs and progeny sex ratio has not been teased apart. Prior to the discovery of *Wolbachia* as a parthenogenesis-inducer, fecundity patterns were suggested to affect the resulting sex ratio in the parasitoid wasp *Muscidifurax uniraptor* (*Legner, 1985*). It was later discovered that *Muscidifurax uniraptor* is a host for parthenogenesis-inducing *Wolbachia*, and that *Wolbachia* titers are positively correlated with the proportion of females produced (*Zchori-Fein, Gottlieb & Coll, 2000*). Here, we use a line of *Trichogramma pretiosum* fixed for *Wolbachia* infection to explore the relationship between patterns of offspring production and sex ratios.

## MATERIALS AND METHODS

### *Trichogramma* colonies

Isofemale lines of *Trichogramma pretiosum* are maintained in 12 × 75 mm glass culture tubes stopped with cotton and incubated at 24 °C, L:D = 16:8. Every 11 days colonies are given honey and egg cards made of irradiated *Ephestia kuehniella* host eggs (Beneficial Insectary, Guelph, Ontario, Canada) afixed to card stock with double-sided tape. Species identification was confirmed by molecular protocols from *Stouthamer et al. (1999)*. We used the "Insectary" line, collected from the Puira Valley of Peru, which has been maintained in a commercial insectary since 1966 (Beneficial Insectary, Guelph, Ontario, Canada). The Insectary line exhibits thelytokous reproduction: females hatch from unfertilized eggs, indicating infection with *Wolbachia*. Infection status was confirmed by PCR following *Werren & Windsor (2000)*.

### Host access experiments

Individual Insectary line wasps from a single generation were isolated during the pupal stage to ensure virginity. Darkened *Ephestia* eggs (indicating a developing *Trichogramma* pupa) were removed from cards using a paintbrush and water, and isolated in 12 × 75 mm glass culture tubes stopped with cotton. Only wasps that hatched singly from an *Ephestia* egg on day one were used in trials, ensuring size-matched, age-matched, virgin wasps. Upon emergence, wasps were subjected to one of four treatments to determine how access to host eggs, and resultant offspring production, affects *Wolbachia* titers and sex ratio (here defined as percentage females among all offspring). Twenty replicate individual wasps

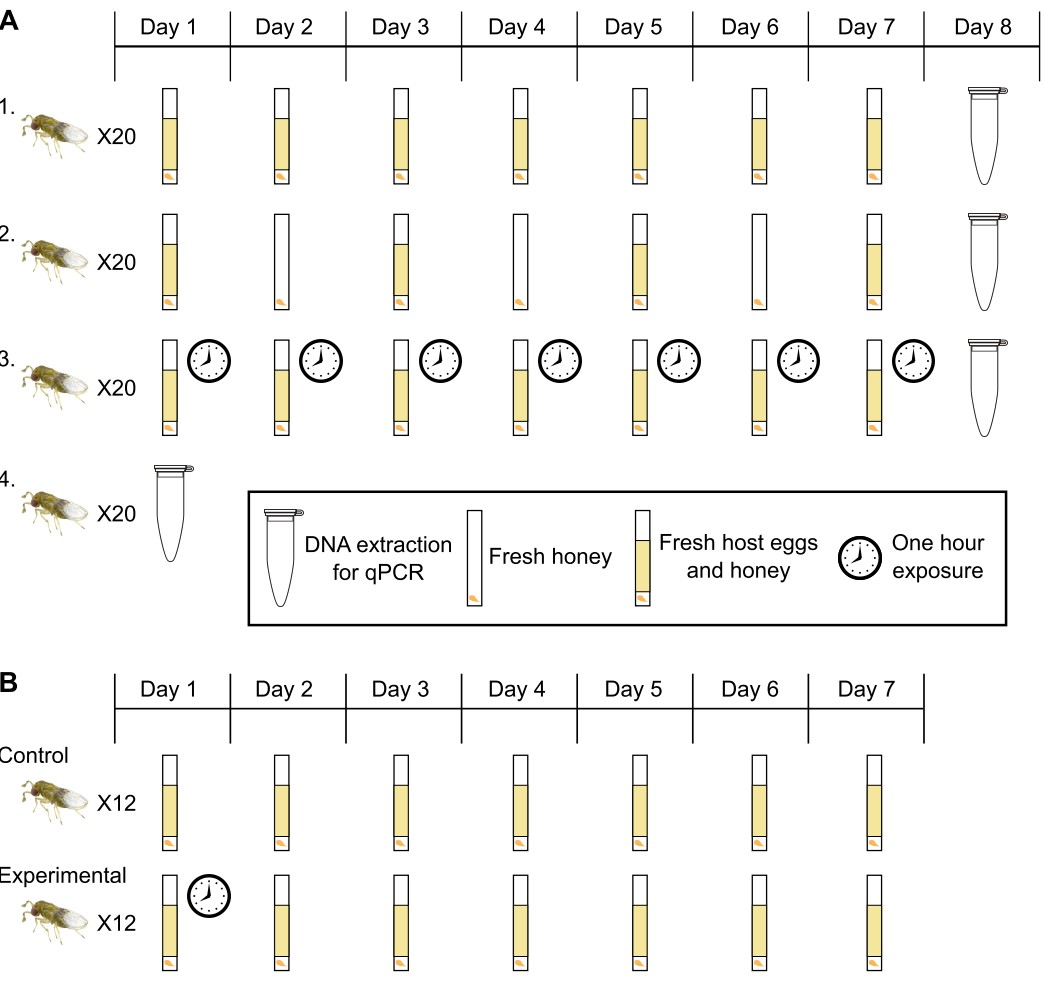

**Figure 1** **Experimental design for host access treatments.** (A) Treatment One: a fresh egg card every 24 h; wasps have constant access to host eggs. Treatment Two: one day on, one day off; wasps have constant access to host eggs every other day. Treatment Three: wasps have access to a fresh egg card for only one hour a day. Treatment Four: collect adult wasps into ethanol immediately upon emergence. (B) Control: a fresh egg card every 24 h; wasps have constant access to host eggs (same as treatment one in 1A). Experimental: wasps have access to a fresh egg card for only one hour on day one, followed by constant access to host eggs starting day two, with a egg cards exchanged every 24 h.

were used for each of the following treatments: (1) a surplus of fresh host eggs every 24 h for seven days, (2) a surplus of fresh host eggs for 24 h every other day, for seven days, (3) a surplus of fresh host eggs for only one hour a day, for seven days, or (4) immediate collection into 100% ethanol upon adult emergence (Fig. 1A). Treatments two and three were designed to restrict access to host eggs, and treatment four was designed to collect data on *Wolbachia* titers in newly emerged wasps. For treatment three, exposure to the fresh egg card was performed at the same time each day, from 10:45 AM–11:45 AM. Egg cards were isolated in individual tubes after the exposure period, ensuring no further parasitization. All mothers, regardless of treatment, were provided with a streak of fresh honey every 24 h. On day eight, all mothers from the first three treatments were collected into 100% ethanol.

**Table 1  Sequences of primers used in this study.**

| Locus | Primer | Sequence (5′–3′) | Amplicon size |
|---|---|---|---|
| 16S | 16S_qF | GAG GAA GGT GGG GAT GAT GTC | 103 bp |
| | 16S_qR | CTT AGG CTT GCG CAC CTT G | |
| *wingless* | wg_qF | AGC TCA AGC CCT ACA ATC CG | 99 bp |
| | wg_qR | CCA GCT TGG GGT TCT TCT CG | |

All offspring from each isolated egg card were allowed to develop, and collected into 100% ethanol within 24 h of adult emergence. Offspring were counted and identified as female, male, or intersex based on antennal morphology. *Wolbachia* quantification (see below) was performed on mothers and select progeny.

## Quantification of *Wolbachia* titers

Total DNA was extracted from wasps using a Chelex method (*Walsh, Metzger & Higuchi, 1991*) as implemented by *Stouthamer et al. (1999)*. Gene sequences from the single-copy *Trichogramma pretiosum* gene *wingless*, and the *Wolbachia* 16S gene were identified from the genome assemblies (GenBank Accession Numbers: JARR00000000 and LKEQ01000000, (*Lindsey et al., 2016*)). *Trichogramma pretiosum wingless* was identified through BLAST searches of the genome, using the *Trichogramma evanescens* homolog as a query (GenBank Accession Number: GQ368153.1). Specific primers (Table 1) were designed to amplify variable regions of these two genes, using primer3 (*Untergasser et al., 2012*). Primer specificity was checked computationally with Primer-BLAST (*Ye et al., 2012*), and against extractions of the moth host eggs, *E. kuehniella*, which has an orthologous copy of *wingless*, and is infected with its own strain of *Wolbachia*. qPCR was performed in 20 μl reactions containing 1× ThermoPol$^{TM}$ buffer (New England Biolabs, Ipswich, MA, USA), 0.4 μM each primer, 200 nM each of dATP, dCTP, and dGTP, 400 nM dUTP, 1 mM MgCl$_2$, 0.5 × EvaGreen$^®$ (Biotium, Fremont, CA, USA), 1 U *Taq* polymerase (New England Biolabs, Ipswich, MA, USA), and 2 μl of sample. Reactions were denatured at 95 °C for 3 min, followed by 35 cycles of 95 °C for 20 s, 58 °C for 20 s, and 72 °C for 20 s. All samples were run in triplicate alongside calibration standards and negative controls on a Rotor-Gene$^®$ Q (QIAGEN). Relative *Wolbachia* titers were determined with the ΔΔCt method (*Livak & Schmittgen, 2001*) with normalization to *wingless*. When testing titers in offspring, we did not correct *wingless* quantification for ploidy levels between males and females as there is evidence that most of the somatic tissues in males are diploid (*Aron et al., 2005*).

## Limiting host access in the first 24 hours

Given the results of the initial host access treatments (Fig. 1A), we set up a second trial to determine the impact of oviposition in the first 24-hour period. Wasps were isolated from a single generation of the Insectary line, and were age and size matched, as before. 12 replicate individual wasps were subjected to either of the following treatments: (1) constant access to fresh host eggs every 24 h (same as treatment 1 in the first experiments), or, (2) one-hour access to an egg card on day one (10:45–11:45AM), followed by constant access to fresh egg cards every 24 h starting day two (Fig. 1B). Trials were carried out for seven

days. Again, mothers received fresh honey every 24 h, and egg cards were isolated after the exposure period. Offspring were allowed to emerge, then counted and identified as female, male, or intersex.

## Statistics

Statistical analyses and data visualization were performed in R version 3.1.2. We used a generalized linear mixed-effects model (GLMM) to assess variation in sex ratios (proportions female and non-female offspring) between treatments using treatment and day of the trial as fixed effects, individual wasp identity as a random effect to account for repeated measures, and a binomial error distribution (*Bates et al., 2014*). We followed up significant effects of treatment with pairwise GLMs, followed by Bonferroni corrections for multiple testing. To assess variation in fecundity among treatments, we used a GLMM with treatment and day of the trial as fixed effects, individual wasp identity as a random effect, and a Poisson error distribution. Here too, we separately assessed variation in total fecundity with a GLM using treatment as a fixed effect, and a Poisson error distribution. We assessed variation in cumulative sex ratio with a GLMM, using cumulative fecundity and treatment as fixed effects, and individual wasp as a random effect. Differences in *Wolbachia* titers between mothers of different host access treatments were assessed with a one-way ANOVA including treatment as a fixed effect, followed by Tukey Honest Significant Difference for post hoc testing. Differences in *Wolbachia* titer between offspring were determined with a linear mixed-effects model, with offspring sex as a fixed effect and mother's identity as a random effect (*Bates et al., 2014*).

## RESULTS

### Sex ratios are more female biased when host access is systematically restricted

To determine how reproductive rate affects sex ratio, we subjected wasps to treatments with different degrees of accessibility to host eggs for seven days. We found a significant effect of the interaction between treatment and day of trial for sex ratio (Fig. 2A; $\chi^2 = 7.38, p = 0.0250$). Daily sex ratio significantly differed by treatment (Fig. 2A; $\chi^2 = 77.38, p < 0.001$) and over time (Fig. 2A; $\chi^2 = 326.07, p < 0.001$). Wasps in treatment three, where access to host eggs was for only one hour a day, consistently produced the most female-biased sex ratios. Treatment one, where wasps had constant access to host eggs, produced the least female-biased sex ratios, with male-bias increasing over time. Wasps subjected to treatment two (hosts every other day) produced intermediate sex ratios. These patterns resulted in overall brood sex ratios that were significantly different between treatments (Fig. 2B; $\chi^2 = 59.72, p < 0.001$).

Similarly, we found a significant effect of the interaction between treatment and day of trial for fecundity (Fig. 2C; $\chi^2 = 181.81, p < 0.001$). Levels of daily fecundity significantly differed by treatment (Fig. 2C; $\chi^2 = 154.65, p < 0.001$), and over time (Fig. 2C; $\chi^2 = 817.83, p < 0.001$). Treatment one wasps produced very high numbers of offspring on day one, and fecundity for the remainder of the trial decayed exponentially. In contrast, reproduction by treatment three wasps was initially lower, but did not drop exponentially

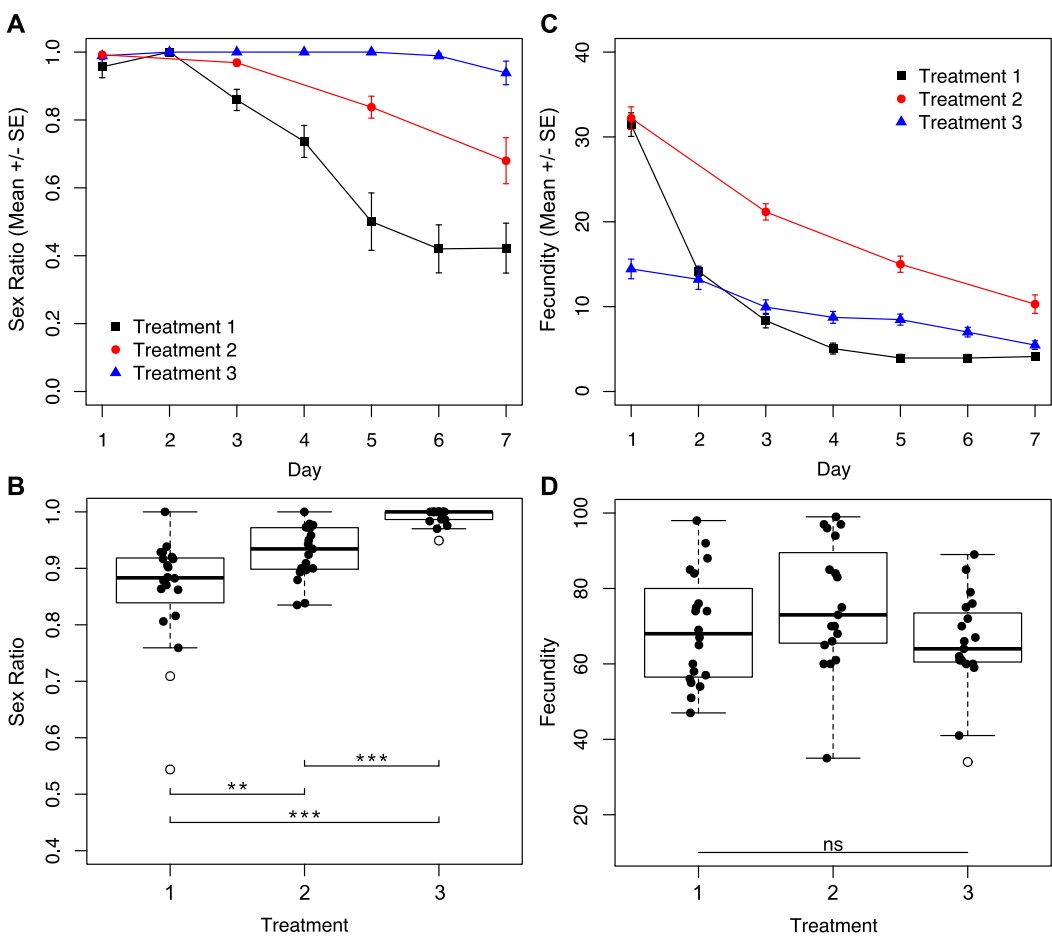

**Figure 2  Sex ratios and fecundity for host access treatments shown in Fig. 1A.** In (B) and (D), open circles represent outliers, double asterisks represent $p \leq 0.01$, and triple asterisks represent $p \leq 0.001$. (A) Temporal variation in sex ratio. (B) Total sex ratio for the seven-day period. (C) Temporal variation in fecundity. (D) Total fecundity for the seven-day period.

as did the reproduction of treatment one wasps. Treatment two wasps had an intermediate level of reproductive output throughout the trial. Unlike total sex ratios, overall fecundity was not significantly different between treatments (Fig. 2D; $\chi^2 = 4.43, p = 0.1091$).

## Periodicity of host access, not cumulative fecundity drives sex ratios

To show that prior offspring production alone was not the driver of sex ratio, we tracked cumulative fecundity and cumulative sex ratios for the duration of the trial, and found a significant effect of treatment on cumulative sex ratio (Fig. 3; $\chi^2 = 38.795, p < 0.001$). The most restrictive treatment (three) results in the production of almost all female offspring. At the same point in total reproductive output, wasps in treatments one and two are producing significantly fewer females (Fig. 3). This shows that it is the restricted access to hosts, and not the total number of offspring produced up to that point that is maintaining the production of more female offspring in treatments two and three.

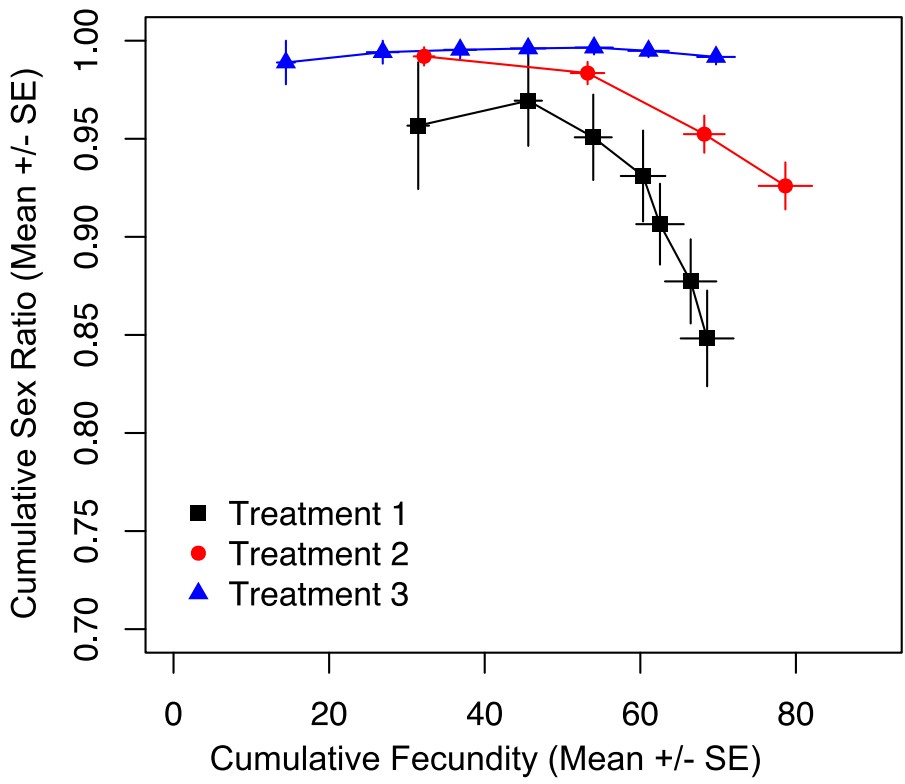

**Figure 3 Cumulative fecundity and sex ratios for host access treatments.** The left-most point of each line represents the first day of the trial, and the right-most point represents the last day of the trial.

## *Wolbachia* titers are higher in wasps that have not recently oviposited

We determined *Wolbachia* titers in mothers from the first four treatments (Fig. 1A), and detected significant differences between treatments (Fig. 4A; $F_{3,70} = 5.559, p = 0.002$). The wasps from treatment four that were collected immediately upon emergence had the highest average *Wolbachia* titers, but they were not significantly different from wasps in treatment three (one hour a day access) ($p = 0.280$). Treatments one and two (constant access, and constant access every other day, respectively) resulted in mothers, on day eight, with significantly lower *Wolbachia* titers relative to immediately collected wasps ($p = 0.033$, and $p = 0.003$ respectively). However, there was no significant difference between treatments one and two ($p = 0.805$), even though egg card access was restricted in treatment two.

## Female wasps have higher *Wolbachia* titers than males

We hypothesized that male offspring would have lower *Wolbachia* titers, as a consequence of having received an insufficient quantity of *Wolbachia* from their mother to induce gamete duplication. We quantified *Wolbachia* titers of three female offspring and three male offspring, from each of three mothers from treatment one (Fig. 1A). After accounting for differences among mothers, *Wolbachia* titer differed significantly between male and female progeny, with female titers that were more than three times higher than in males (Fig. 4B; $\chi^2 = 22.22, p < 0.001$). With the exception of one male, all female offspring had higher titers than their male siblings.

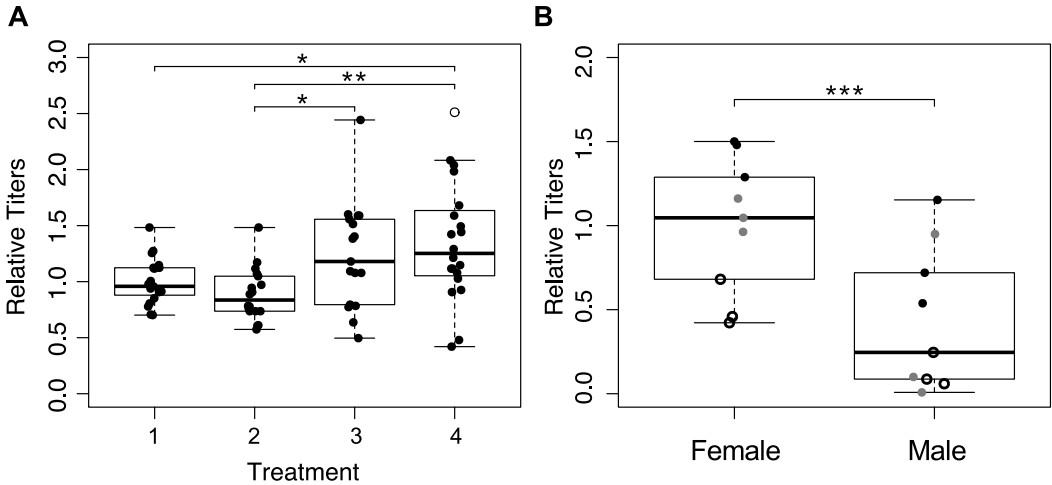

**Figure 4** *Wolbachia* **titers, as measured by quantification of 16S relative to *wingless*.** Within a plot, titers have been normalized to the sample shown most left. Open circles represent outliers, a single asterisk represents $p \leq 0.05$ and double asterisks represent $p \leq 0.01$. (A) *Wolbachia* titers of mothers collected after the host access treatments one through four. Only significant pairwise comparisons are denoted. (B) *Wolbachia* titers of the offspring produced by mothers subjected to treatment one. Point styles denote offspring that originated from the same mother.

## Reproductive output in the first 24-hours significantly affects sex ratios after one week

Given the finding that the largest difference in fecundity between treatments one and three was during the first 24 h, we set up a second set of experiments in which one group of wasps' access to egg cards was restricted only on day one, and a second group of wasps had constant access to host eggs (Fig. 1B). Under these conditions, we found no interactive effect of treatment and day on sex ratio (Fig. 5A; $\chi^2 = 1.90, p = 0.1677$). However, we found a significant effect of treatment (Fig. 5A; $\chi^2 = 8.75, p = 0.0031$), and a significant effect of day (Fig. 5A; $\chi^2 = 201.46, p < 0.001$) on sex ratio. The experimental treatment (only one hour with an egg card on day one) maintained higher female-biased sex ratios for the duration of the trial. The overall sex ratio of the experimental treatment was significantly more female-biased than the control (Fig. 5B; $\chi^2 = 24.40 p < 0.001$). In contrast to the sex-ratio results for these trials, we found a significant interactive effect of treatment and day on fecundity (Fig. 5C; $\chi^2 = 7.79, p = 0.0053$), as well as a significant effect of day alone (Fig. 5C; $\chi^2 = 395.74, p < 0.001$). Importantly, there was no significant difference in total fecundity between treatments (Fig. 5D; $\chi^2 = 1.83, p = 0.1761$).

## DISCUSSION

Based on the established relationship between *Wolbachia* titers and the parthenogenesis-phenotype (*Pascal et al., 2004*; *Stouthamer, 1997*; *Tulgetske & Stouthamer, 2012*; *Zchori-Fein, Gottlieb & Coll, 2000*), and previous research on *Muscidifurax uniraptor* that showed sex ratios changed with reproductive patterns (*Legner, 1985*), we hypothesized that reproductive rate might mediate the level of male production in an asexual line of *Trichogramma*. Restricted access to hosts is likely the more ecologically relevant condition,

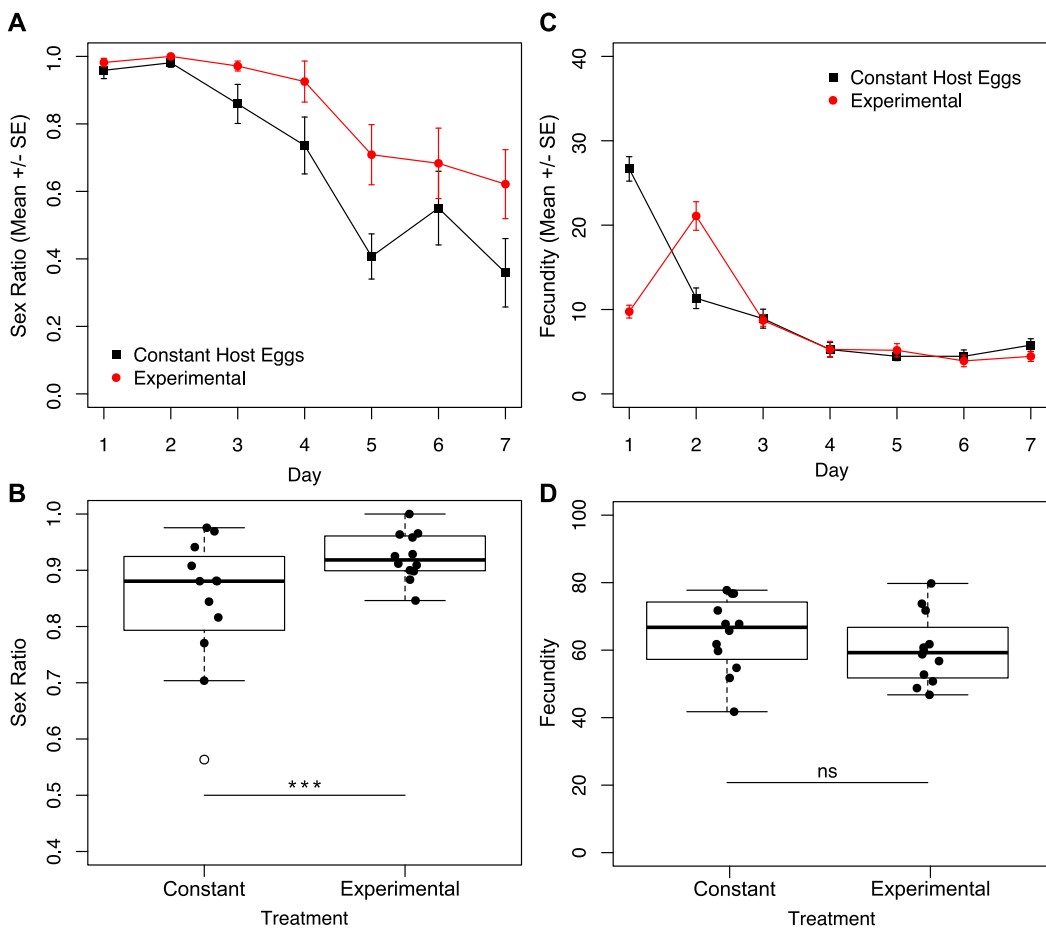

**Figure 5  Sex ratios and fecundity for host access treatments shown in Fig. 1B.** In (B) and (D), open circles represent outliers and triple asterisks represent $p \leq 0.001$. (A) Temporal variation in sex ratio. (B) Total sex ratio for the seven-day period. (C) Temporal variation in fecundity. (D) Total fecundity for the seven-day period.

so the males produced under high host availability conditions in the lab would not be produced under field conditions. In natural settings, host availability is often patchy and limited: fluctuations in environmental conditions and the requirement to physically relocate to find suitable host eggs pose barriers to constant oviposition. Through manipulation of *Trichogramma* oviposition rates, by limiting access to host eggs, we saw that patterns of offspring production had a significant effect on total sex ratio. When wasps were not able to parasitize host eggs continuously, either by alternating days with access to eggs, or limiting the time per day with egg access, sex ratios were maintained at higher levels (Fig. 2A). In fact, for wasps that had access to host eggs for only one hour a day, the near-complete parthenogenesis phenotype was maintained for the duration of the trial, without significant impact on total fecundity (Fig. 2D). Critically, it is only in the first 24 h where treatment one wasps (constant host eggs) show drastically different fecundity than the treatment three wasps (host eggs for one hour a day). On day two, mothers of these two treatments produced nearly the same number of offspring, and for the remainder of

the trial the treatment three wasps produced higher numbers of offspring (Fig. 2C). High fecundity within the first 24 h had a lasting effect on the sex ratio of progeny produced for the remainder of the trial.

We show that it is not cumulative fecundity alone that determines the likelihood of the next offspring being feminized (Fig. 3). This corroborates the finding that there is no significant difference in total fecundity between treatments. We see that sex ratios drop precipitously in treatment one when approximately 45 offspring had been produced, significantly diverging from the host-limited treatments. Even restricting access to hosts on only the first day has a prolonged effect on the sex ratio of the offspring (Fig. 5A).

Results from qPCR analysis of *Wolbachia* titers were mixed. It is worth noting that whole-body extractions, which are necessary for the minute *Trichogramma*, likely do not provide the most resolved look at *Wolbachia* titers in the ovaries, which could be responsible for symbiont provisioning to the egg (*Ferree et al., 2005*). Regardless, *Wolbachia* titers were highest in immediately collected wasps, which is congruent with our expectation (Fig. 4A). The most restrictive egg card access treatment maintained *Wolbachia* titers at a level comparable to those of wasps who had yet to reproduce, indicating that *Wolbachia* titers had been sustained (Fig. 4A). However, treatment two, which produced intermediate sex ratios, resulted in *Wolbachia* titers that were indistinguishable from treatment one wasps that oviposited constantly, albeit significantly lower than the immediately collected and treatment three wasps (Fig. 4A). We predict that this is reflective of the fact that wasps from both of those treatments were able to oviposit up until their collection; whereas mothers from treatment three had 23 h of recovery prior to collection, resulting in *Wolbachia* titers similar to those that had yet to oviposit.

Wasps that oviposit relatively constantly may be maintaining a rate of germ line stem cell turnover such that *Wolbachia* cannot keep up, resulting in fewer *Wolbachia* deposited into eggs, and the subsequent production of males. This is consistent with the finding that a large number of *Wolbachia* are transferred from nurse cells into the developing oocyte (*Ferree et al., 2005*; *Serbus et al., 2008*). Alternatively, *Wolbachia* from somatic tissues could continually invade the germ line to ensure a high proportion of infected eggs (*Fast et al., 2011*; *Toomey et al., 2013*). Under these circumstances, constant oviposition may draw *Wolbachia* from the soma resulting in lower whole-body titers. We propose that the recovery periods built into our host access treatments are critical to maintaining *Wolbachia* titers high enough in developing eggs to ensure effective parthenogenesis induction. This is also in line with previous studies that showed a positive relationship between *Wolbachia* titers and sex ratios in parthenogenesis-inducing *Wolbachia* (*Pascal et al., 2004*; *Stouthamer & Mak, 2002*; *Zchori-Fein, Gottlieb & Coll, 2000*).

Additional support for the hypothesis that *Wolbachia* titer is essential for parthenogenesis induction comes from finding of lower *Wolbachia* titers in males compared to their sisters (Fig. 4B). While it is possible that adult titers may not be reflective of the number of *Wolbachia* deposited into the egg, arguably, our data point to titers being important for proper parthenogenesis-induction, which is again consistent with previous findings on the effect of heat and antibiotics (*Pascal et al., 2004*; *Pintureau, Chapelle & Delobel, 1999*; *Stouthamer, Luck & Hamilton, 1990*), and the importance of maternal loading of

*Wolbachia* into eggs (*Ferree et al., 2005*; *Serbus et al., 2008*). There is a chance that some of the phenotypic males with higher *Wolbachia* titer could be of female karyotype, which has been shown to occur in related *Trichogramma* species and other parthenogenesis-inducing *Wolbachia* infected wasps (*Ma et al., 2015*; *Tulgetske, 2010*). We would expect these individuals to have high enough *Wolbachia* titers to induce gamete duplication, but not high enough to result in the hypothesized epigenetic feminization that occurs afterward (*Tulgetske, 2010*).

It is likely that *Wolbachia* titers in the egg may not be the final determinant of successful parthenogenesis induction, but instead it is a *Wolbachia*''-secreted factor that needs to be at sufficient levels. This has been hypothesized as a mechanism for the sex-ratio changes in *Muscidifurax* (*Zchori-Fein, Gottlieb & Coll, 2000*), and is the mechanism for inducing cytoplasmic incompatibility, as sperm do not contain *Wolbachia* cells, but have been modified by *Wolbachia*-derived proteins (*Beckmann, Ronau & Hochstrasser, 2017*; *LePage et al., 2017*). Females from other closely related species of *Trichogramma* hatch with a set of fully developed eggs, but will mature new eggs over the course of their adult life (*Volkoff & Daumal, 1994*). The newly matured eggs may need a longer ''incubation time'' in order to accumulate the appropriate concentration of *Wolbachia* or a *Wolbachia*-derived parthenogenesis factor. More resolved studies of *Wolbachia* densities, *Wolbachia*-protein densities, and the time that eggs spend in the mother, would aid in identifying a threshold level of infection critical for effective parthenogenesis induction.

There is evidence for gene flow between populations of *Trichogramma* in the field, and that *Wolbachia*-infected females can mate with males and fertilize their eggs (*Stouthamer & Kazmer, 1994*). Given that access to host egg resources has an impact on the likelihood of males being produced, the amount of gene flow may fluctuate with environmental conditions. While limited availability of host eggs is likely the norm, lepidopteran populations fluctuate, with abundance peaking during certain seasons or in response to particular weather patterns (*Kunte, 1997*; *Pollard, 1988*; *Roy et al., 2001*; *Van den Bosch, 2003*). Environmental conditions could have direct effects on *Wolbachia* titers (such as high temperatures decreasing bacterial titers (*Pintureau et al., 2002*; *Stouthamer, Luck & Hamilton, 1990*)), and indirect effects through availability of host eggs. Greater host availability may lead to an increase in offspring production, and if high enough, a decrease in sex ratio. Males produced under these circumstances would likely provide a mechanism for gene flow between asexual lineages.

The higher penetrance of parthenogenesis induction under host-limited conditions, as occurred in our study can in part explain the frequent coexistence of infected and uninfected females in *Trichogramma* field populations (*Huigens et al., 2004*; *Stouthamer, 1997*; *Stouthamer, Luck & Hamilton, 1990*). How these populations can coexist has been unclear, because laboratory experiments with infected and uninfected lines from these field populations often showed that under constant host availability (such as in our treatment one), the daughter production of infected females was lower than that of mated uninfected females (*Silva et al., 2000*; *Stouthamer & Luck, 1993*).

In conclusion, we provide evidence for *Trichogramma* reproductive patterns mediating the parthenogenesis phenotype, likely through the depletion of *Wolbachia* titers. The males

produced during times of high oviposition rates may provide an opportunity for gene flow between populations, and thus new host-symbiont combinations. Given the interest in using *Wolbachia* as a tool to control insect populations (*Hoffmann, Ross & Rasic, 2015*; *Turelli & Hoffmann, 1991*), it is critical that we understand the context-dependent nature of *Wolbachia* phenotypes, and how this may result in different selective pressures for the host-symbiont relationship.

## ACKNOWLEDGEMENTS

We thank Barbara Baker and Christina Luu for their assistance in collecting many tiny wasps into ethanol, and Sarah Lillian for her statistical advice. We also thank Paul Rugman-Jones, Eric Smith, Matthew Daugherty, Jason Stajich, and three reviewers for helpful discussions and feedback on drafts of the manuscript.

### Funding

This work was supported by the National Science Foundation (DEB 1501227 to ARIL); the United States Department of Agriculture (NIFA 194617 to RS and NIFA 2016-67011-24778 to ARIL); and Robert and Peggy van den Bosch Memorial Scholarships to ARIL. The funders had no role in study design, data collection and analysis, decision to publish, or preparation of the manuscript.

### Grant Disclosures

The following grant information was disclosed by the authors:
National Science Foundation: DEB 1501227.
United States Department of Agriculture: NIFA 194617, NIFA 2016-67011-24778.

### Competing Interests

The authors declare there are no competing interests.

### Author Contributions

- Amelia R.I. Lindsey conceived and designed the experiments, performed the experiments, analyzed the data, contributed reagents/materials/analysis tools, wrote the paper, prepared figures and/or tables, reviewed drafts of the paper.
- Richard Stouthamer conceived and designed the experiments, contributed reagents/materials/analysis tools, wrote the paper, reviewed drafts of the paper.

### Supplemental Information

Supplemental information for this article can be found online at http://dx.doi.org/10.7717/peerj.3505#supplemental-information.

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
