# Peer review of "Penetrance of symbiont-mediated parthenogenesis is driven by reproductive rate in a parasitoid wasp"

_PeerJ, doi:10.7717/peerj.3505_

## Round 0.1 · original submission · Minor Revisions

· Academic Editor

Minor Revisions

Dear Drs. Lindsey and Stouthamer:

I have now received three reviews of your manuscript “Penetrance of symbiont-mediated parthenogenesis is driven by reproductive rate in a parasitoid wasp”. I am happy to convey to you that all three reviews were very favorable. Congratulations! Before we can officially accept your work for publication in PeerJ, there are suggestions from all three reviewers that I would like to see you entertain. In particular, Reviewer 2 has made substantial comments and suggestions…please take these seriously. Please address Reviewer 1’s concerns with the statistical analysis. Also, please modify Figure 1 as suggested by Reviewer 3. All of the reviewers’ concerns should be addressed in your response letter, and I believe that this will greatly improve your manuscript.

Good luck with your revision, I look forward to seeing it.

-Joe

·

Basic reporting

A well-written contribution, with nice figures.

Some minor comments:
line 42-46. Explanation of CI not relevant for this paper

line 96. "improve female-biased sex ratios" sounds strange. Why is a more female-biased sex ratio "better"? More neutral wording would be better.

line 147. "each" should be "either"

The Tulgetske GM. 2010. reference is incomplete.

Experimental design

It would have been nice to see the depletion in time, by determining titres at several timepoints for each of the treatments.

Statistics: I am not familiar with the permutational multivariate approach taken. Please add a sentence explaining its advantage over the more traditional binomial GLM. Does it take brood size into account when assessing sex ratio?

Validity of the findings

Line 205-207 and 223. It is good practise to discuss significant interaction effects first, before interpreting the effects of individual factors.

line 225-228. This statement is not supported by the lack of an interaction effect on sex ratio (line 220).

Additional comments

A well-conducted study that strongly indicates that the level of PI is determined by Wolbachia titre and that Wolbachia titre is affected by reproductive output. It would be interesting to see follow-up work that characterizes these naturally produced males further in terms of fertility, ploidy and perhaps any latent ability to induce CI.

Reviewer 2 ·

Basic reporting

Specific suggestions for improving clarity, context and referencing are provided in attached review.

Experimental design

no comment.

Validity of the findings

The authors are encouraged to speculate more, as doing so is well-substantiated by existing context of the literature, and will further connect readers to the full implications of the study. (Specific suggestions provided in attached review.)

Additional comments

Please see attached review.

Annotated reviews are not available for download in order to protect the identity of reviewers who chose to remain anonymous.

Reviewer 3 ·

Basic reporting

The manuscript is well written, the figures are easily interpreted, and the statistical analysis and reporting are complete.

Experimental design

The research question is well defined and the experiments are designed to test the hypothesis that restriction of host eggs for oviposition will sustain female-based sex ratios.

Validity of the findings

The data is robust with adequate replication. The discussion assesses each result with adequate depth, and acknowledges alternative arguments which were not addressed directly in the experiments. The authors discuss the ecological significance of the results, and propose mechanisms which may result in effective parthenogenesis induction.

Additional comments

I would suggest adding a separate experimental design figure for the second trial or amending figure 1 to include the second treatment of the second trial. Figure 1 is well designed and a similar figure for the second trial will aid in clarity.

---

## Round 0.2 · accepted · Accept

· Academic Editor

Accept

Congratulations on the acceptance of your manuscript! It will be a great contribution to the field. Great job on the revision. Thanks for being patient with the review process, and for choosing to publish with PeerJ. I hope we will see more of your work submitted to us in the future.

Best,

-joe